# Computational development of rubromycin-based lead compounds for HIV-1 reverse transcriptase inhibition

Carlos E.P. Bernardo and Pedro J. Silva

REQUIMTE/Faculdade de Ciências da Saúde, Universidade Fernando Pessoa,
Rua Carlos da Maia, Porto, Portugal

## ABSTRACT

The binding of several rubromycin-based ligands to HIV1-reverse transcriptase was analyzed using molecular docking and molecular dynamics simulations. MM-PBSA analysis and examination of the trajectories allowed the identification of several promising compounds with predicted high affinity towards reverse transcriptase mutants which have proven resistant to current drugs. Important insights on the complex interplay of factors determining the ability of ligands to selectively target each mutant have been obtained.

## INTRODUCTION

HIV reverse transcriptases are multifunctional enzymes which use the virus single-stranded RNA genome as template to build a double-stranded DNA which may later be incorporated into the host's genome. They are composed of two subunits: p66 acts both as a DNA polymerase and as a RNAase which cleaves RNA/DNA hybrid molecules and p51 (whose sequence is equal to that of p66, but lacks the last 124 aminoacids) plays mostly a structural role. Due to its crucial role in the virus life cycle, HIV reverse transcriptase (RT) has been the target of several successful drug-developing efforts. These drugs may be grouped in several classes based on their mechanism of action (thoroughly reviewed in *Jochmans, 2008*; *Sarafianos et al., 2009*; *Singh et al., 2010*): nucleoside analogue RT inhibitor (NRTI), like azidothymidine (*Mitsuya et al., 1985*) (the first successful drug against HIV) act as an alternative substrate and block the synthesis of the viral DNA due to their lack of a free 3' OH- group; nucleotide-competing RT inhibitors (NcRTI) like INDOPY-1 (*Jochmans et al., 2006*) bind the active site in an as-yet-undisclosed manner; and non-nucleoside RT inhibitors (NNRTI) in contrast bind to the enzyme in a hydrophobic pocket 10 Å away from the active site (*Kohlstaedt et al., 1992*; *Ding et al., 1998*) and prevent the enzyme from attaining a catalytically competent conformation. Since reverse transcriptases lack a proofreading ability, very high rates of mutation are observed and mutants resistant to one or more drugs frequently arise. To decrease the probability of selection of drug-resistant strains, a combination therapy including drugs with different targets and modes of action is most often used in clinical practice. Still, newer drugs must be continually developed to fight resistant strains.

Corresponding author
Pedro J. Silva, pedros@ufp.edu.pt

Rubromycins are a small class of compounds containing naphtoquinone and 8-hydroxyisocoumarin moieties (*Brasholz et al., 2007*). In 1990, $\beta$- and $\gamma$-rubromycin were shown to inhibit HIV-1 reverse transcriptase (*Goldman et al., 1990*), although at levels that were also toxic to human T lymphocytes. $\gamma$-rubromycin was later shown to be an inhibitor of human telomerase (*Ueno et al., 2000*), fueling interest in its use as an anti-cancer agent. The development of less toxic variants of these lead compounds has long been prevented due to the difficulty of their laboratory synthesis, but several synthetic routes to these interesting molecules have recently become available (*Akai et al., 2007*; *Rathwell et al., 2009*; *Wu, Mercado & Pettus, 2011*; *Wilsdorf & Reissig, 2014*), enabling the evaluation of many simpler derivatives as candidates for the inhibition of telomerase (*Yuen et al., 2013*). As far as we could ascertain, no derivatives of $\gamma$-rubromycin with substitution patterns as complex as those observed in the natural molecule have yet been synthesized. As we envisage that such derivatives might afford higher selectivity towards selected reverse-transcriptase mutants or more favorable pharmacokinetic properties, we decided to evaluated several not-yet-synthesized $\gamma$-rubromycin derivatives using computational docking and molecular dynamics simulations of the most promising candidates. The results are compared to those of the commercially-available, 2nd-generation NNRTI drug rilpivirine.

## COMPUTATIONAL METHODS

All computations were performed in YASARA (*Krieger et al., 2004*) using the crystal structure of the rilpivirine-inhibited HIV1 reverse transcriptase published by Das et al. (PDB: 2ZD1) (*Das et al., 2008*). A double-mutant structure, (p66)K103N/(p66)Y181C and a quadruple mutant (p51p66)M184I/(p51p66)E138K, were also generated to evaluate the robustness of the ligand binding to reverse transcriptase variants with increased resistance to NNRTIs: K103N is known to strongly reduce susceptibility to efavirenz and nevirapine (*Bacheler et al., 2001*; *Rhee et al., 2004*; *Eshleman et al., 2006*; *Zhang et al., 2007*; *Melikian et al., 2014*) and E138K has a similar effect towards rilpivirine, which is increased by M184I (*Kulkarni et al., 2012*); Y181C reduces suceptibility to efavirenz, etravirine and rilpivirine (*Reuman et al., 2010*; *Tambuyzer et al., 2010*; *Rimsky et al., 2012*). $\gamma$-Rubromycin-based ligands (Fig. 1 and Supplemental Information 1) were docked to the wild-type structure with AutoDock 4.2.3 (*Morris et al., 2009*) using default docking parameters and point charges assigned according to the AMBER03 force field (*Duan et al., 2003*). The highest scoring ligands and poses were selected for molecular dynamics simulations. Initial structures for molecular dynamics simulations of mutant proteins were generated from the corresponding ligand-bound wild-type structures through mutation of the corresponding aminoacids. All simulations were run with the AMBER03 forcefield (*Duan et al., 2003*), using a multiple time step of 1.25 fs for intramolecular and 2.5 fs for intermolecular forces. Simulations were performed in cells 5 Å larger than the solute along each axis (final cell dimensions $127.3 \times 102.6 \times 78.8$ Å), and counter-ions (88 $Cl^-$ and 77 $Na^+$) were added to a final concentration of 0.9% NaCl. In total, the simulation contained approximately 106,500 atoms. A 7.86 Å cutoff was taken for Lennard-Jones

**Figure 1 Structures of the tested γ-rubromycin-based ligands.** Substitution patterns in molecules 1–14; 17; 20–21 and 26–46 are shown in Supplemental Information 2 and Supplemental Information 1.
forces and the direct space portion of the electrostatic forces, which were calculated using the Particle Mesh Ewald method (*Essmann et al., 1995*) with a grid spacing <1 Å, 4th order B-splines and a tolerance of $10^{-4}$ for the direct space sum. Simulated annealing minimizations started at 298 K, velocities were scaled down with 0.9 every ten steps for a total time of 5 ps. After annealing, simulations were run at 298 K. Temperature was adjusted using a Berendsen thermostat (*Berendsen et al., 1984*) based on the time-averaged temperature, i.e., to minimize the impact of temperature control, velocities were rescaled only about every 100 simulation steps, whenever the average of the last 100 measured temperatures converged. Substrate parameterization was performed with the AM1BCC protocol (*Jakalian et al., 2000*; *Jakalian, Jack & Bayly, 2002*). All simulations were run for 30 ns. Differences in ligand binding energies between wild-type and mutant proteins were evaluated using the MM-PBSA methodology (*Srinivasan et al., 1998*). Although MM-PBSA is unable to afford accurate absolute binding energies and the high standard deviation of MM-PBSA energies limits its ability to discriminate between ligands with similar binding-affinities (*Weis et al., 2006*) to a protein, its application to the analysis of the affinity of a single molecule to a series of protein mutants affords high quality results (*Moreira, Fernandes & Ramos, 2007*; *Martins et al., 2013*), presumably due to better cancellation of errors (as the effect of a point-mutation on a large protein is, in relative terms, much smaller than that of a substitution in a small molecule). For each snapshot (taken at 0.25 ns intervals from the last 15 ns of the simulation) we computed the molecular mechanics energy of the protein–ligand complex, the electrostatic contribution to solvation energy (using the Adaptive Poisson–Boltzmann Solver; *Baker et al., 2001*) and nonelectrostatic contributions to solvation (with a surface-area-dependent term; *Wang et al., 2001*). These computations were repeated for each snapshot for the ligand-free protein and the protein-free ligand, to obtain an estimate of the average binding energy of each ligand. Normal mode analysis computations were performed using the Webnm@ server at http://apps.cbu.uib.no/webnma/home (*Hollup, Salensminde & Reuter, 2005*).

## RESULTS

Computational docking allows the fast screening of a large number of candidate ligands, which may afterwards be analyzed through more demanding computational techniques in the search for suitable leads for further development and experimental characterization. Out initial screen analyzed the docking performance of $\gamma$-rubromycin derivatives with/without truncated rings, substitution of the oxygen atoms appended to the spirocyclic ring and different substitution patterns around the rings. Twenty-six of the tested $\gamma$-rubromycin-based ligands bind preferentially to an exposed pocket in subunit p51 formed by the Glu89-His96 loop and the Pro157-Leu187 helix-turn-sheet motif. This pocket lies very far away from the nucleic acid binding surface (Fig. 2), which completely prevents this binding mode from competitively inhibiting the reaction mechanism.

This distant binding pocket might still affect the catalytic activity of the enzyme by triggering a conformational change from the active "open" conformation (*Ding et al., 1998*) to an inactive conformation. Since such transitions are usually too slow to be

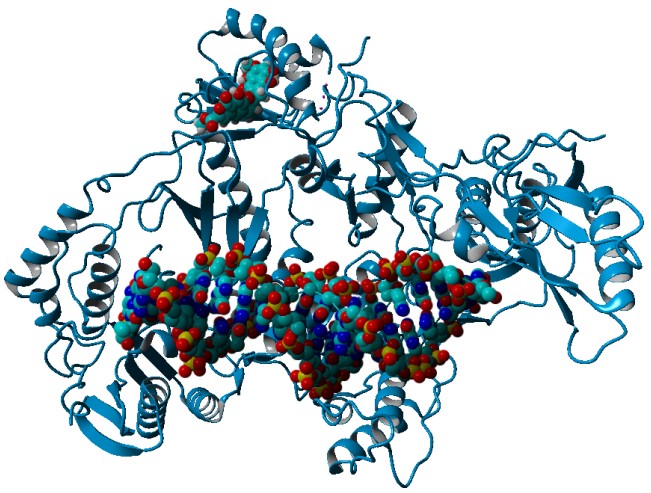

**Figure 2** **Preferential binding mode of ligand 28 to HIV reverse transcriptase, as computed with AutoDock, with superposed DNA molecule taken from the DNA-bound HIV structure (PDB: 2HMI)** (*Ding et al., 1998*).

observed with molecular dynamics simulations, we analyzed the available vibrational modes of HIV-1 reverse transcriptase using the efficient algorithm and simplified force field described by *Hinsen (1998)*. In this method, the protein is simulated as a coarse-grained series of springs connecting every C$\alpha$ with all other C$\alpha$ atoms with exponentially-decaying force-constants. Despite its conceptual simplicity, the computed vibrational modes and vibration frequencies have been shown to correlate very well with those observed in explicit molecular dynamics simulations. Furthermore, important conformational changes can most frequently be explained by the first few non-trivial vibrational modes, which enables its use in the location of allosteric transitions (*Tama & Sanejouand, 2001*; *Zheng & Brooks, 2005*; *Zheng, Brooks & Thirumalai, 2006*; *Zheng, Brooks & Thirumalai, 2007*; *Rodgers et al., 2013*; *Sanejouand, 2013*). Application of this method to the catalytically active "open" conformation of HIV-1 reverse transcriptase (PDB: 2HMI) (*Ding et al., 1998*) shows that inclusion of a coarse-grained representation of $\gamma$-rubromycin in the proposed binding site does not affect the protein flexibility: indeed, hardly any changes in vibrational modes are observed, as confirmed by the very high correlation coefficients between the normal modes of ligand-bound and empty reverse transcriptase, which always exceeding 0.9977. Figure 3 shows the contributions of each aminoacid to the first six non-translational, non-rotational modes obtained by this method, and clearly highlights the negligible contribution of the aminoacids lining this proposed binding pocket to the overall flexibility of the enzyme.

Several $\gamma$-rubromycin-based ligands (**11**, **12**, **18**, **21**, **30**, **31**, **33–46**) may bind the previously defined NNRTI-binding pocket with affinities exceeding those of this distant, inactive, binding pocket. The most promising leads (Table 1) generally had (like the NNRTI drug rilpivirine) a nitrile group appended to the ligand. The behavior of these molecules in the reverse transcriptase binding pocket of wild-type and mutant reverse transcriptase was then evaluated through 30 ns-long molecular dynamics simulations

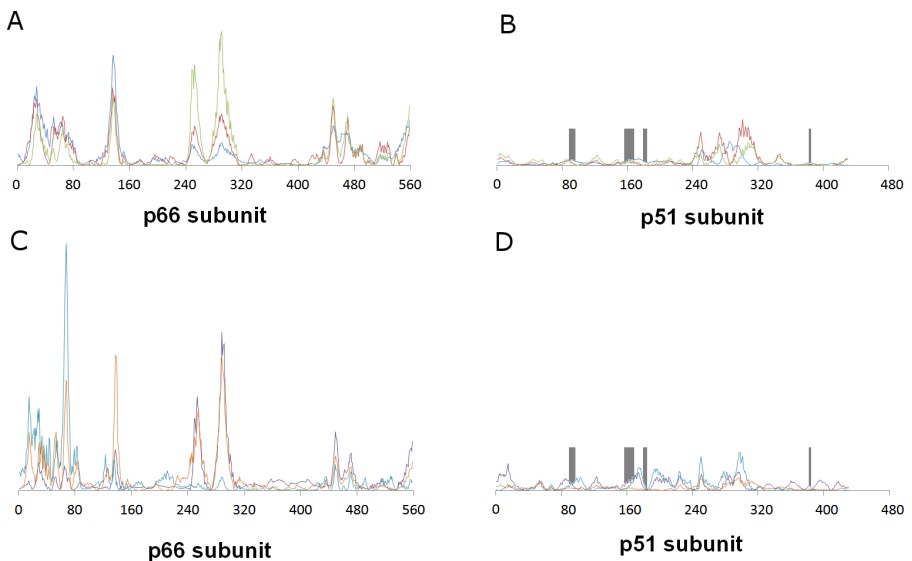

**Figure 3 Relative contribution of each amino acid displacement to the first six non-trivial normal modes of HIV-1 reverse transcriptase.** (A) and (B) Modes 7 (blue), 8 (red) and 9 (green). C and (D) Modes 10 (violet), 11 (light blue) and 12 (orange). The regions lining the proposed binding pocket are highlighted in dark gray.

**Table 1 Substitution patterns and AutoDock-computed binding energies of the best-scoring ligands to the previously characterized NNRTI binding pocket.** Only differences from the parent compound ($\gamma$-rubromycin) are shown. The binding energy of the drug rilpivirine, computed with the same methods, amounts to $-13.25$ kcal mol$^{-1}$. Data for all ligands is available as Supplemental Information 2.

| Ligand: | $\gamma$-rubromycin | 46 | 36 | 27 | 45 | 13 | 38 | 37 |
|---|---|---|---|---|---|---|---|---|
| $R_1=$ | $-COOCH_3$ | | | | $-CH_2OH$ | | | |
| $R_2=$ | $-C=C-H$ | | | (S) HC–CH$_2$ | | | | |
| $R_3=$ | $-C=O-O$ | | | | $-O-C=O$ | | | |
| $R_4=$ | $=C=C-H$ | $-C-CH_2-$ | | | | | | |
| $R_5=$ | $-C-OH$ | $-C=O$ | | | | | | |
| $R_6=$ | $-O-$ | | | | | | | |
| $R_7=$ | $-O-$ | | | | | | | |
| $R_8=$ | $-OH$ | | | | | | | |
| $R_9=$ | $-OH$ | | | | | | | |
| $R_{10}=$ | $-C=O$   $-C=O$ | | | | | | | |
| $R_{11}=$ | $-O-CH_3$ | $-CN$ | $-F$ | | $-CN$ | $-CH_2-CH_3$ | $-CN$ | $-Cl$ |
| Binding energy | $-12.95$ | $-13.71$ | $-13.72$ | $-13.82$ | $-13.82$ | $-13.91$ | $-14.25$ | $-14.29$ |

and compared to that of rilpivirine. The worst-scoring ligands towards the NNRTI binding pocket were those where any of the rings had been removed, as well as the ones where the oxygen at the $R_6$ position was substituted by nitrogen or carbon. Surprisingly, substitution of the $=CH-$ at the $R_4$ position by an isoelectric $=N-$ (ligand **32**) also led to a dramatic loss of binding affinity. Binding affinities of each ligand to wild-type and mutant HIV-1 RT s were computed with the MM-PBSA methodology using the last

**Table 2 Binding affinity (average ± standard error of the mean) of the best-scoring ligands to reverse-transcriptase mutants, relative to the binding affinity of each ligand to the wild-type enzyme. Values in kcal mol[1].** Negative values show stronger binding than observed to the wild-type protein.

| | K103N / Y181C | E138K / M184I |
|---|---|---|
| Rilpivirine | 1.6 ± 0.9 | 3.6 ± 0.8 |
| γ-rubromycin | 9.8 ± 1.1 | 0.3 ± 1.1 |
| 13 | −6.7 ± 1.4 | −16.8 ± 1.4 |
| 27 | 7.7 ± 1.4 | −13.0 ± 1.2 |
| 36 | 10.3 ± 1.0 | −7.1 ± 0.9 |
| 37 | −4.0 ± 1.2 | −5.2 ± 1.4 |
| 38 | 4.6 ± 1.0 | −4.1 ± 0.9 |
| 45 | −3.6 ± 1.0 | −7.0 ± 1.2 |
| 46 | −1.9 ± 1.2 | −3.8 ± 1.0 |

15 ns of each molecular dynamics simulation (Table 2). This method, while not accurate enough to produce reliable absolute binding free energies, has been shown to provide good estimates of binding affinity trends provided that either the ligands or the protein targets under comparison are very similar (*Massova & Kollman, 2000*). The computed data for rilpivirine agree with the experimentally observed sensitivity of its binding to E138K/M184I variants, and to the relative insensitivity of its effect on the presence/absence of K103N or Y181C mutation, which supports the applicability of the MM-PBSA approach to this system. Ligands **13**, **27**, **36** and **45** are computed to bind significantly stronger to the rilpivirine-resistant E138K/M184I HIV1-RT variant than to the wild-type protein, and may therefore be suitable lead compounds for further pharmaceutical developments against rilpivirine-resistant strains. Further insight to the determinants of binding affinity was obtained through close inspection of each simulation.

As observed in the crystal structure (*Das et al., 2008*), rilpivirine remains bound to RT throughout the simulation through a large number of hydrophobic contacts and two very stable hydrogen bonds with the backbone of Lys101, whether in the wild-type or any of the tested mutants. Its high hydrophobicity strongly favor it to adopt a very buried conformation and low solvent-accessible area throughout the simulation. The high stability of the hydrogen bonds does not change in the mutated variants, but the total number of close hydrophobic contacts between rilpivirine and the protein does become smaller in the E138K/M184I mutant, which is consistent with the experimentally observed lower affinity of this drug towards it (*Singh et al., 2012*), and the computed MM-PBSA binding energy.

γ-rubromycin is a much larger and less flexible ligand than rilpivirine: as it binds to the NNRTI binding patch, the methoxy-bearing end of γ-rubromycin remains in contact with the solvent through its hydrophilic surface (Fig. 4), whereas the oxygen atoms in its naphtoquinone moiety establish stable hydrogen bonds with Lys101 and Lys103. In the K103N/Y181C mutant, γ-rubromycin becomes less exposed to the solvent, since the shorter sidechain of Asn103 (compared to the wild-type Lys 103) forces the naphtoquinone

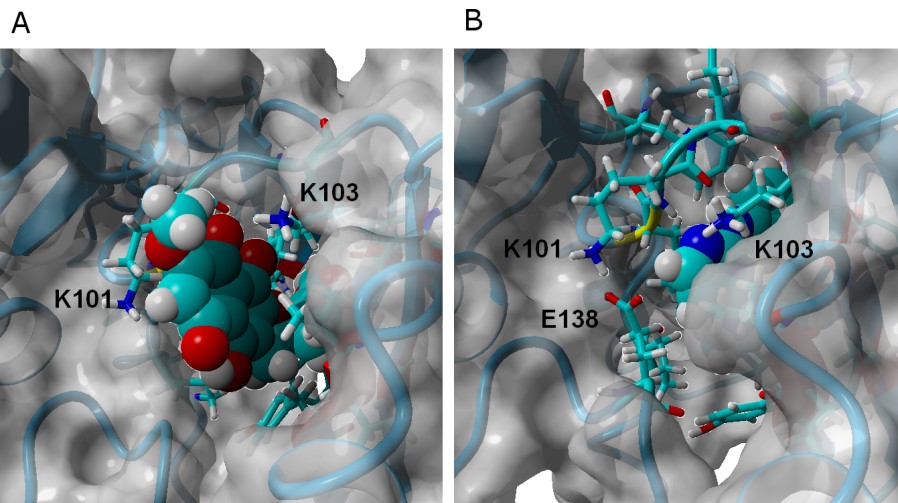

**Figure 4  γ-rubromycin (A) and rilpivirine (B) bound to wild-type HIV-1 reverse transcriptase.** Snapshots were taken from random points in the last 15 ns of molecular dynamics simulations.

moiety of the ligand to penetrate deeper into the crevice in order to establish a stabilizing hydrogen bond with Asn103. The buried conformation of γ-rubromycin removes the methoxy group from its favored solvent-exposed environment leading to a binding mode which is computed by MM-PBSA to be markedly less favored than observed in the wild-type protein, but which remains stable due to the difficulty in breaking the large number of favorable hydrogen bonds to Asn103 and Lys101. γ-rubromycin binding to the E138K/M184I is very similar to the wild-type protein: hydrogen bonds between the ligand and Lys101 and Lys103 are also present (though ca. 0.4 Å longer), and subtle cavity rearrangements due to the loss of the ionic bridge between Lys101 and (p51)Glu138 (which is mutated to a Lys) lead to the possibility of intermitent H-bonded interactions between the carbonyl of Ile180 (or the sidechain of (p51)Thr139) and the naphtoquinone moiety.

The binding of ligand **13** to wild type RT differs more from that of γ-rubromycin than would be expected from the very small difference in their structures (the single substitution of a methoxy group in γ-rubromycin by an ethyl): since the ethyl group is less hydrophilic than a methoxy, it initially tends to establish a hydrophobic interaction with the sidechain of Val179, instead of protruding (like the methoxy group) in the direction of the solvent, leading to a binding mode where the stabilizing hydrogen-bonds between the ligand and the protein are due to Glu138 instead of Lys101. In contrast to what is observed in the binding of γ-rubromycin to the K103N/Y181C, the replacement of the Lys-based H-bonds does not lead to an unfavorable buried conformation of the ligand because, as the simulation progresses, the interaction with Glu138 causes subtle changes in the local environment which becomes more exposed to the solvent than originally: indeed, there is in average one more water molecule near ligand **13** than near γ-rubromycin, leading to a smaller desolvation penalty when **13** binds to the protein. Binding of **13** to the mutants is strongly favored over binding to the wild-type due to the formation of hydrogen bonding to the backbone of Ile180 (especially in E138K/M184I) and especially by the changes in the

electrostatic component of ligand solvation caused by the presence of two intra-molecular H-bonds in **13** when bound to the mutant proteins.

The sp³ hybridization in the acetyl-bearing carbon of the isocoumarin-moiety in **27** introduces a deviation from full planarity in that region of the ligand, which facilitates its interactions with the Trp229 and Ty188 aminoacids on that end of the NNRTI-binding cavity. Ligand **27** is found to bind much more favorably to the quadruple mutant E138K/M184I (with a very large number of very short and stable hydrogen bonds with Lys101, Lys103, Lys138 and Thr139) than to wild-type or K103N/Y181C, where the only stable hydrogen bonds available are those with Glu138. The electrostatic component of the solvation energy of **27** follows the opposite trend as the protein is changed from WT to the mutants, but the smaller variation of this factor simply dampens the magnitude of the change in binding affinities brought about by the variation in protein–ligand interactions.

Ligand **36** bears a fluorine atom in place of the methoxy group carried by $\gamma$-rubromycin. Like ligand **27**, **36** has higher affinity to the E138K/M184I mutant than to either the wild-type and, especially, the K103N/Y181C mutant. The minute size of the fluorine substituent allows Lys101 and Glu138 (which lie on opposite sides of the crevice where the ligands bind) to approach each other and form a strong ionic bridge which pushes the ligand further inside the cavity. This ionic bridge cannot form in the E138K/M184I mutant, leading to a binding mode where the ligand is slightly more exposed and strongly binds to Lys101, Lys103 and Lys 138. In the K103N/Y181C, the interactions between ligand and protein are weaker due to the strong deviations from 180° in the possible H-bonding partners in the binding cavity.

Ligands **37** and **38** bear a chlorine and a cyanide (respectively) in place of the fluorine present on **36**. The intermediate size of these substituents (relative to the fluorine in **36** and the methoxy in $\gamma$-rubromycin) leads to an intermediate degree of penetration in the binding cavity, between those of **36** and in $\gamma$-rubromycin. As observed in most cases, Lys101 is responsible for the most stable interaction between protein and ligand. No single contribution is, however, determinant in the observed trend of binding affinity of **37** to the proteins, as the correlation of total binding energies to either electrostatic components of solvation or to protein–ligand interaction is insignificant: the overall effect is rather the result of subtle interplay of the electrostatic component of solvation and the protein–ligand interaction. Solvation effects, in contrast are determinant in the binding trends observed for ligand **38**, as the higher affinity to the M184I/E138K mutant is correlated to its much smaller desolvation penalty, which is due to the considerable exposure of its nitrile group to solvent when the entrance to the binding channel is not blocked by the Glu138-Lys101 ionic bond (Fig. 5).

Ligand **45** bears, like ligands **38** and **46**, a nitrile group in the position occupied by a methoxy in $\gamma$-rubromycin. It differs from **38** by the replacement of the acetyl substituent of the isocoumarin by a hydroxymethyl and by switching the orientation of the lactone group in isocoumarin from –O–C=O to O=C–O. The replacement of acetyl from hydroxymethyl makes the isocoumarin end of **45** significantly smaller and less hydrophilic, leading that end of the molecule towards the inside of the crevice and the nitrile-bearing

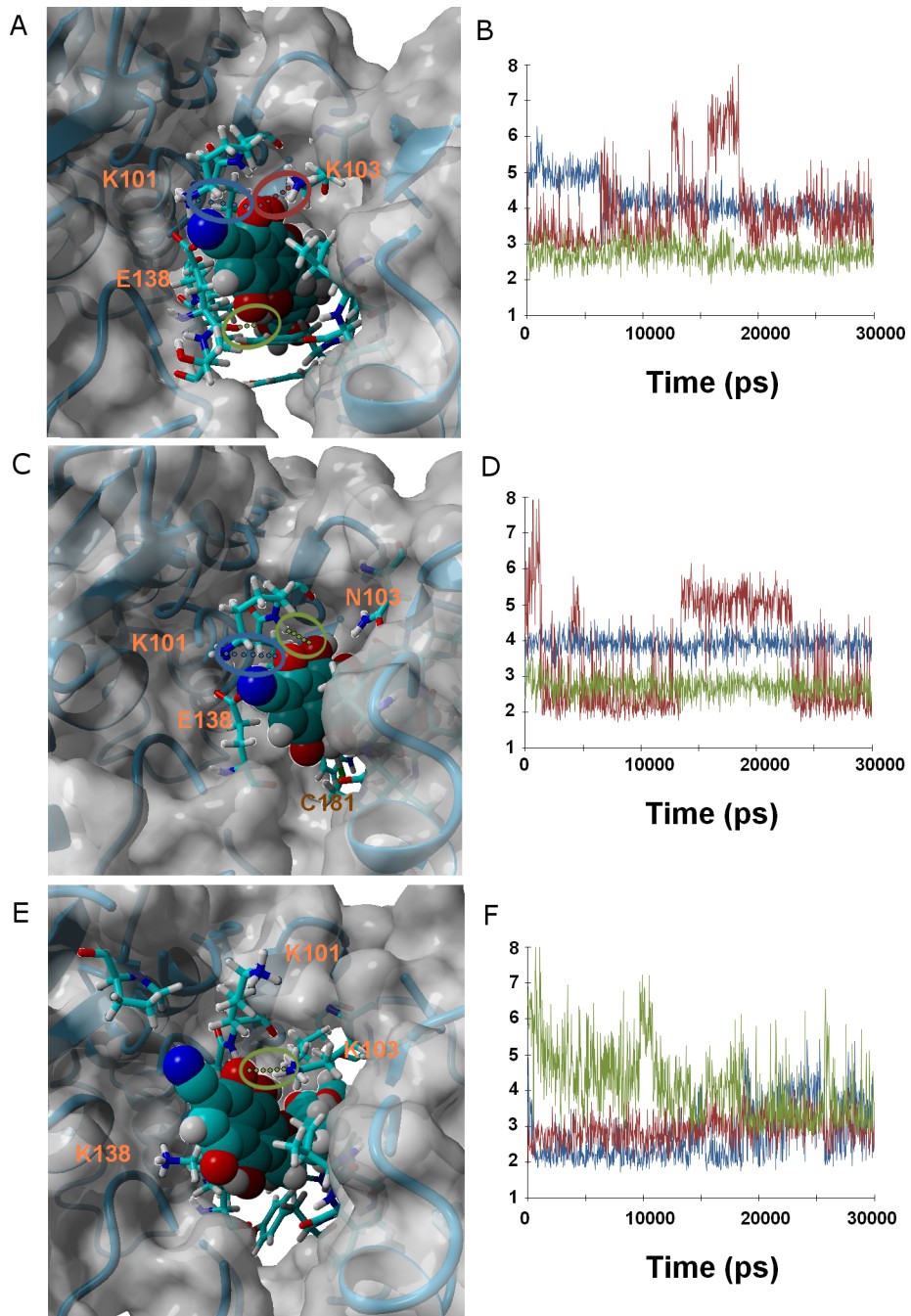

**Figure 5 Ligand 38 bound to wild-type (A and B), K103N/Y181C (C and D) and E138K/M184I (E and F).** Snapshots were taken from random points in the last 15 ns of molecular dynamics simulations. As far as possible, the bonds depicted in the graphs have been highlighted with the same color in the corresponding image (the exceptions are the C181 ••• OH hydrogen bond shown as the red line in D, and the two hydrogen bonds between K101 backbone NH and nearby oxygen atoms in the inhibitor shown as red and blue lines in F).

naphtoquinone portion of **45** to protrude from the other end of the cavity into the solvent. The only H-bonds between ligand and protein now involve the backbone atoms of Lys101 and Glu138. These H-bonds weaken considerably in both mutants, but this destabilizing effect is overtaken by sizable stabilizing effects due to favorable solvation, leading to overall better binding to K103N/Y181C and (especially) E138K/M184I.

Other than the lack of H-bond donating ability in its isocoumarin moiety (due to the replacement of its hydroxyl by a carbonyl), ligand **46** is identical to ligand **38**. Unlike ligand **38**, its ability to bind the K103N/Y181C mutant is not inferior to its affinity to the wild-type protein: the presence of a carbonyl instead of a hydroxyl allows it to accept a hydrogen bond from Tyr183, which is able to rotate into position in the mutant due to the smaller size occupied by Cys181 (compared to the original tyrosine present in the wild type.)

We also performed molecular dynamics simulation of ligands **32** and **24** bound to the NNRTI-binding pocket, as these ligands are ranked by AutoDock among the weakest in affinity towards this binding site. These simulations are expected to shed light into structural factors that disfavor binding to HIV-1 reverse transcriptase.

Ligand **24** (like the weakly-binding ligands **22**, **23** and **26**) lacks the spirocyclic scaffold characteristic of rubromycins. The longer head-to-tail distance due to the opening of the central acetal leads to a binding mode where the naphtoquinone moiety is buried inside the enzyme in a position similar to that taken by isocoumarin in the other simulations and the isocoumarin end becomes much more exposed to the solvent than observed for naphtoquinone in the binding modes of the other ligands. This portion of the molecule now remains continuously exposed to the solvent, significantly decreasing the number of stable hydrophobic ligand-protein interactions. (Figures 6A and 6B.)

Ligand **32** differs from $\gamma$-rubromycin only in the replacement of the =CH group at the $R_4$ position by a nitrogen atom. This small isoelectronic change does not, however, lead to very dramatic differences in dynamic behavior, as evidenced by the high similarity of the resulting simulation to that of the original $\gamma$-rubromycin: Lys103 still maintains a strong and stable interaction to the naphtoquinone moiety, and Lys101 now interacts with the appended methoxy group in $R_{11}$, rather than the –OH group in $R_8$ (Fig. 6). The weaker binding of **32** predicted by the docking analysis seems to be due mostly to the cost of burying the polar =N– group away from the solvent, rather than on any large differences of ligand-protein interactions.

## INFLUENCE OF LIGAND BINDING ON PROTEIN DYNAMICS

In the absence of NNRTI, reverse transcriptase may adopt either a compact structure (*Hsiou et al., 1996*) or an open structure (*Ding et al., 1998*) which allows the binding of a RNA template and the polymerization of DNA (Fig. 7). The binding of an NNRTI acts as a "wedge" (*Ivetac & McCammon, 2009*) that further separates the catalytic triad (Asp110, Asp185 and Asp186) from Met230, which is believed to be part of the primer-recognition region. The same "wedge" effect is observed for all tested ligands bound to the NNRTI

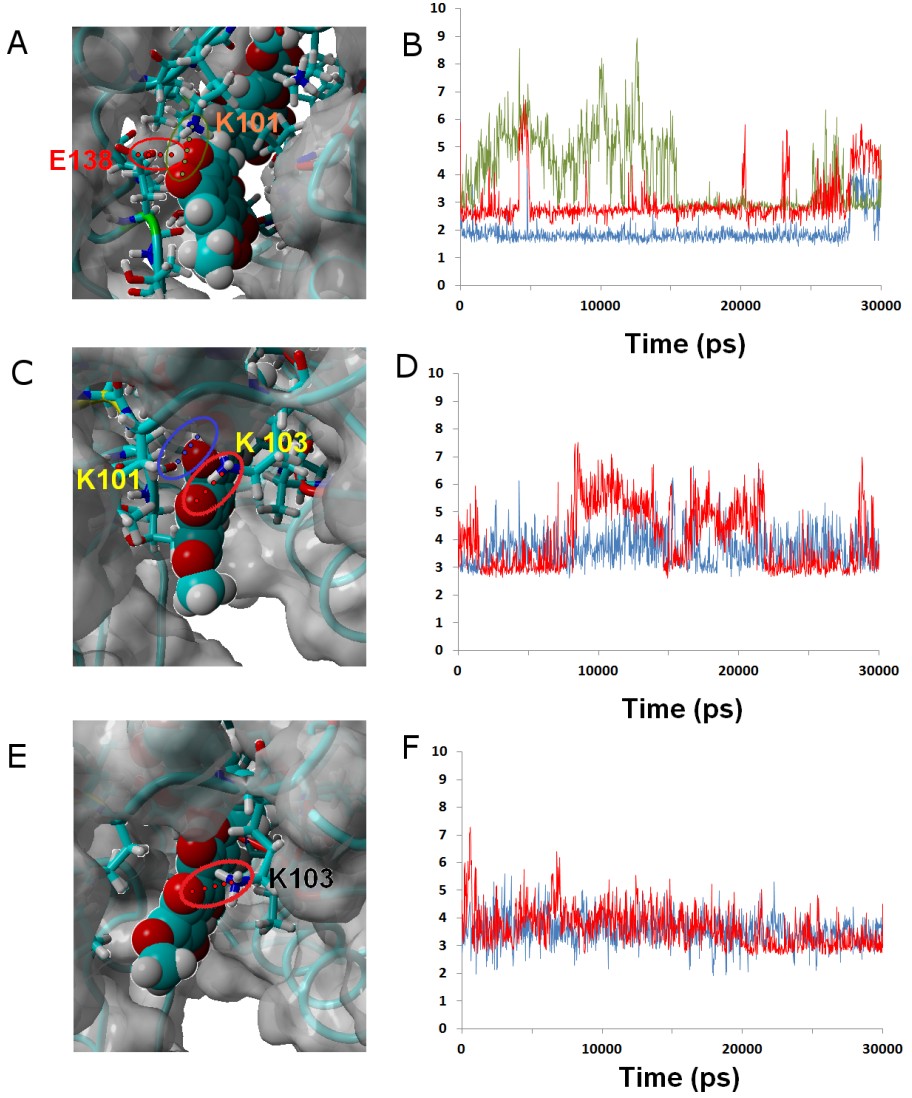

**Figure 6 Ligands 24 (A and B), 32 (C and D) and $\gamma$-rubromycin (E and F) bound to wild-type reverse transcriptase.** Snapshots were taken from random points in the last 15 ns of molecular dynamics simulations. As far as possible, the bonds depicted in the graphs have been highlighted with the same color in the corresponding image (the exceptions are the I180 carbonyl ● ● ● OH hydrogen bond shown as the blue line in the B, and the hydrogen bond between K101 carbonyl and methoxy oxygen in $\gamma$-rubromycin shown as a blue line in F).

binding pocket (Fig. 7 and Supplemental Information 6–16). The only instance where the diagnostic Lys154-Asn255 and Glu67-Gln242 distances stably assume values as short as those observed in the catalytically active conformation is observed in the simulation of the K103N/Y181C mutant in the presence of ligand **46**.

## CONCLUSIONS

Our computational study confirms that $\gamma$-rubromycin-based ligands are able to bind to HIV1-reverse transcriptase at the previously defined NNRTI-binding site, and

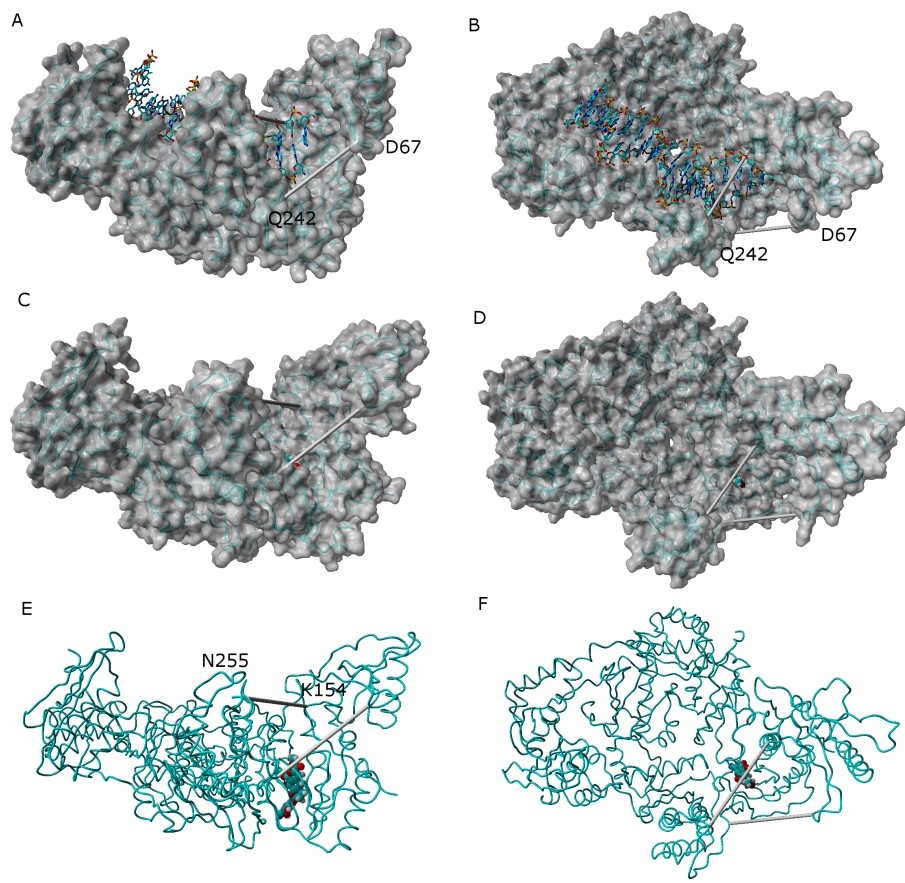

**Figure 7 Comparison between the catalytically active "open" conformation of HIV-1 reverse transcriptase and the "extended" conformations obtained in the simulations.** Side-view (A) and "top"-view (B) of nucleic acid-bound reverse transcriptase (PDB: 2HMI) (*Ding et al., 1998*). Side-view (C and E) and "top"-view (D and F) of a typical ligand-bound structure obtained in our simulations. The highlighted distances can be used as "fingerprints" for the identification of catalytically competent conformations.

allowed the identification of ligands that are predicted to bind very strongly to RT mutants which have shown high resistance towards other NNRTI compounds. The best compounds (**13**, **27**, **36** and **45**) achieve selective binding to the highly resistant mutant E138K/M184I through very subtle variations on the degree of exposure to solvent, and on the number and strength of hydrogen bonds and hydrophobic interactions with the protein. These molecules should therefore become good candidates in the quest for suitable $\gamma$-rubromycin-based drugs.

### Funding

Research at REQUIMTE is supported by Fundação para a Ciência e a Tecnologia through grant no. PEst-C/EQB/LA0006/2011. This work has been financed by FEDER through Programa Operacional Factores de Competitividade–COMPETE and by Portuguese Funds through FCT–Fundação para a Ciência e a Tecnologia under project PTDC/QUI-

QUI/111288/2009. The funders had no role in study design, data collection and analysis, decision to publish, or preparation of the manuscript.

### Grant Disclosures

The following grant information was disclosed by the authors:

Fundação para a Ciência e a Tecnologia: PEst-C/EQB/LA0006/2011.

Programa Operacional Factores de Competitividade–COMPETE.

FCT–Fundação para a Ciência e a Tecnologia: PTDC/QUI-QUI/111288/2009.

### Competing Interests

The authors declare there are no competing interests.

### Author Contributions

- Carlos E.P. Bernardo performed the experiments, analyzed the data, prepared figures and/or tables, reviewed drafts of the paper.
- Pedro J. Silva conceived and designed the experiments, performed the experiments, analyzed the data, contributed reagents/materials/analysis tools, wrote the paper, prepared figures and/or tables, reviewed drafts of the paper.

### Data Deposition

The following information was supplied regarding the deposition of related data:

Complete analysis files have been deposited in Figshare:

Pedro J. Silva (2014): Molecular dynamics of rubromycin derivatives bound to WT and mutant reverse transcriptase: http://dx.doi.org/10.6084/m9.figshare.979429.

### Supplemental Information

Supplemental information for this article can be found online at http://dx.doi.org/10.7717/peerj.470.

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
