# Peer review of "Computational development of rubromycin-based lead compounds for HIV-1 reverse transcriptase inhibition"

_PeerJ, doi:10.7717/peerj.470_

## Round 0.1 · original submission · Major Revisions

· Academic Editor

Major Revisions

The reviewers have found your manuscript to be written clearly, however they consider representation, methodology and analyses to be too speculative. It is important that you address their concerns carefully as your revised manuscript will require re-review.

Reviewer 1 ·

Basic reporting

The article is clearly written and the introduction is outlined well. However the authors fall short in providing clear figures and also a more extensive discussion of the results is needed.

For example, in Figure 1 is is not clear at first sight what structures 1-14, 17, 20-21, 26-46 actually represent, please refer to the table in supplementary information and emphasize (using a reference) where, how and why these compounds were synthesized. To my surprise, there is also not a structure of rilvipirine included, which would greatly enhance the interpretation of the results. The nomenclature is also not always clear, sometimes rubromycin is mentioned and sometimes, gamma-rubromycin. Please clarify to avoid confusion between both forms or just use gamma-rubromycin in all instances.

In addition, the figures of the docked conformations are not comprehensible. The figures only display ligands on top of the surface of the protein, which is not informative to understand the interactions with the protein amino acids. For example, the authors refer to hydrogen bond with Asn103, and also discuss a Glu138-Lys101 ionic bond in Figure 3, but both are absent in the figures. Another concern is that snapshots were taken from random points and therefore are possibly not representative. I advise the authors to either show a structure at the end of the simulation or use a structure after conformational clustering. Also, Figure 4 is way too complex to comprehend. It is unclear which amino acids surround the ligand and what the surface represents. I suggest to directly indicate the isocoumarin and naphtoquinone parts of the ligand (using for example arrows or colouring).

Finally, I would suggest to fuse table 1 and 2 together (using actual binding free energy values and not just relative values as used in table 2). To make the interpretation for the reader more straightforward, it would help to add a figure to visually explain the binding free energy differences from table 2.

Experimental design

One of the caveats of the current method is that all conformations were docked into a cavity that was induced by binding of rilvipirine (PDB 2ZD1). While I understand the necessity of the authors to have a suitable docking site in reverse transcriptase, they must acknowledge that this is a rather biased starting point. After their docking procedure, they only relax the binding poses of the best compounds. Hence, I would suggest to also include some of the bad binding inhibitors for MD and MM/PBSA analysis. Currently, the selection could be misleading and preferred to already rilvipirine induced sites.

Validity of the findings

- A more detailed discussion/conclusion is missing. Which compounds do the researchers suggest for further optimization and why (compared to rilvipirine)?
- The authors do not correlate with any experimental information of the efficacy of the inhibitors. The authors should look into the literature or at least mention if no experimental data is available.

Reviewer 2 ·

Basic reporting

The manuscript is well written and easy to follow
however,
The figures are of a poor quality. They are very unclear. please represent the the proteins in cartoon, if using surfaces please make them transparant as now half of the compounds are invisibible.

Experimental design

I have some serious doubts about the validaty of the experimental design.
The authors investigate the binding mode of Rubromycin to HIV-RT. In the introduction it is already mentioned that they also inhibit telomerase. Furthermore these compounds inhibit other polymerases.(Mizushina, et al 2000) The biochemical mode of inhibition of DNA polymerase beta by alpha-rubromycin.

These compounds were found to be direct DNA substrate competitors or interact with DNA. These proteins do not have an NNRTI binding side, but all of them have a DNA interacting surface ... It is unclear on what ground the authors have decided that these compounds would bind to the NNRTI pocket in HIV-RT. This should at least be added to the manuscript.
As a remark. The insensitivity towards certain resistance mutations can also be explained by binding to a different binding site.

Furthermore the compounds do not exhibit any similarity with previous inhibitors. In fact the overall shape of the compounds is very different from Rilpivirine. Please indicate why the Rilpivirine structure was choosen and not one of the many other HIV-RT inhibitors complexes. As this pocket is very flexibile it addapts to the ligand. Therefore, in the case the compounds truly would work by binding to the pocket it is important to select the correct conformation of the binding receptor, as this will influence the docking results.
It would have been better to choose different receptor conformations and dock in all of them instead of just one conformation, analyse these results before continuing the results.

Validity of the findings

The authors have not shown that they have produced stables simulations (no run-away ligands , stable RMSD during the simulation)

The authors have performed MM/PBSA simulations, this produces a binding affinity which should correlate with the inhibitory portency (IC50 or Ki) in case of a correct simulation. There is no such figure in the manuscript.

And all results depend on hypothesis that these compounds act by binding into the NNRTI pocket which remains unvalidated.

---

## Round 0.2 · accepted · Accept

· Academic Editor

Accept

The manuscript has been amended to most major concerns from the reviewers. It does require careful language editing.